# Determination of the Assigned Values of Blood Cells by an Impedance Method for Hematological Reference Samples Used in Hematology External Quality Assessment (EQA) Programs

**DOI:** 10.3390/biomedicines10123169

**Published:** 2022-12-07

**Authors:** Ngoc Nguyen Vo, Huu Tam Tran, Thi Hong Phuong Nguyen, Dinh Dung Vu, Thanh Sang Vo, Thuy Huong Nguyen

**Affiliations:** 1Department of Biotechnology, Faculty of Chemical Engineering, Ho Chi Minh City University of Technology (HCMUT), 268 Ly Thuong Kiet Street, District 10, Ho Chi Minh City 700000, Vietnam; 2Vietnam National University Ho Chi Minh City, Linh Trung Ward, Thu Duc District, Ho Chi Minh City 700000, Vietnam; 3Center for Standardization and Quality Control in Medical Lab of HCMC, 75A Cao Thang Street, District 3, Ho Chi Minh City 700000, Vietnam; 4Institute of Applied Technology and Sustainable Development, Nguyen Tat Thanh University, Ho Chi Minh City 700000, Vietnam

**Keywords:** hematological reference samples, pseudo-leucocytes, pseudo-platelets, EQA, assigned value, hematology testing, blood cell, RBC, WBC, PLT

## Abstract

The research into and production of hematological reference samples used to implement an external quality assessment (EQA) to check the quality of hematology tests are necessary for hematology laboratories in Vietnam. In this research, the study team determined the assessment values of blood cell count (human RBCs, pseudo-leucocytes, and pseudo-platelets) by the impedance method used in hematology EQA programs. The hematological reference samples were controlled at three concentration levels: low, normal, and high. Determination of the assigned value (mean ± 2SD) was performed for the following hematology analyzer series by impedance method: ABX Micros 60, Celldyn 1700, and Mindray BC 2000. Each device was sent to 10 different laboratories for evaluation. Research results for assigned values of each model (ABX Micros 60, Celldyn 1700, and Mindray BC 2000) were determined at the three concentrations. For the ABX Micros 60 and Celldyn 1700 series, 80% of laboratories had analytical results within assigned values. For the Mindray BC 2000 series, 100% of laboratories had analytical results within assigned values. The measurement results for the number of human RBCs, pseudo-leucocytes, and pseudo-platelets on each analyzer were similar between the 10 laboratories; the results of the three hematology analyzer series using the impedance method were different and the difference was statistically significant (*p* < 0.05). Thus, hematological reference samples for measuring the number of blood cells meeting the standards so that they can evaluate the results of laboratories using the impedance method: ABX, Celldyn 1700, Mindray BC 2000.

## 1. Introduction

Hematology testing is one of the most basic and important areas of laboratory testing. Hematology tests contribute to the diagnosis and treatment of blood diseases. Therefore, hematology testing requires high reliability and quality control using tools such as internal quality control (IQC) and external quality assessment (EQA) [1,2,3].

The research and production of hematological reference samples has been performed for a long time worldwide. Kim et al. studied hematological reference samples using human and animal red blood cells (RBCs) to replace white blood cells (WBCs) and platelets (PLTs) with shark, crocodiles, turkey, or samples that were readable on automatic hematology analyzers for the impedance and optical methods of Abbott and Streck [4,5,6]. In Japan, Kawai et al. also produced hematology controls using fresh blood from patients and performed an EQA program on WBCs in nine laboratories. However, the sample had to be analyzed within 8 h of sample collection [7]. In Vietnam, there is currently no hematological reference sample suitably designed for the clinical situation of the laboratories; all hematological reference samples used in EQA programs are imported from abroad. Therefore, the research into and production of hematological reference samples help laboratories in Vietnam proactively source samples and reduce the cost of purchasing samples. In previous research, we have fixed human RBCs, used goose and goat RBCs to replace human WBCs (pseudo-leucocytes), and used human PLTs (pseudo-platelets). We have determined the optimal storage medium formulas for the three types of red blood cells and the sample shelf-life was determined to be 3 months [8]. In this research, we performed threshold determination for three concentrations levels (high, normal, and low) in hematology analysis. Subsequently, we determined the assigned valued and evaluated the assigned value of the following hematology analyzers by an impedance method: ABX Micros 60, Celldyn 1700, and Mindray BC 2000.

## 2. Materials and Methods

### 2.1. Materials

Human RBCs (from the blood bank at HCMC Hematology and Transfusion Hospital). Goose RBCs (*Anser cygnoides*, 5–6 months age) have been used to make stimulated pseudo-leucocytes because of their similarity in size to human WBCs and having a nucleus. Goat RBCs (*Capra aegagrus hircus*, 3–4 months age) have been used to make stimulated pseudo-platelets because of their similarity in size to human PLTs. It is simple to collect blood from animals in Vietnam. Neomycin sulfate, chloramphenicol, sodium azide, cell stabilization buffer, glycerol, formaldehyde, and glutaraldehyde were purchased from Sigma-Aldrich (Burling, MA, USA). Automatic hematology test equipment: ABX Micros 60 (Horiba, Longjumeau, France), Celldyn 1700 (Abbott, Abbott Park, IL, USA), and Mindray BC 2000 (Nanshan District, Shenzhen, China).

### 2.2. Determination of Threshold Values for Three Levels of High, Normal, and Low Concentrations of Human RBCs, Pseudo-Leucocytes, and Pseudo-Platelets

We used clinical thresholds to determine the thresholds for normal values of RBCs, WBCs, and PLTs. Based on the analytical measurement range or the linearity range of the three analyzers (ABX Micros 60, Celldyn 1700, and Mindray BC 2000), pathological values for the upper and lower bounds were determined, from which we determined the low and high threshold values of human RBCs, pseudo-leucocytes, and pseudo-platelets.

### 2.3. Evaluation of the Assigned Values

After fixing, human RBCs, pseudo-leucocytes, and pseudo-platelets, were combined and the number of blood cells was measured on the ABX Micros 60 of Center for Standardization and Quality Control in Medical Laboratory of Ho Chi Minh City (CSQL of HCMC) laboratory. At low concentrations (HH1), the human RBCs value is from 0.23–3.49 × 10^12^/L, pseudo-leucocytes count is from 0.8–4.49 × 10^9^/L, and pseudo-platelets count is from 10–149 × 10^9^/L; at normal concentrations (HH2), the human RBCs value is from 3.5–5.5 × 10^12^/L, pseudo-leucocytes is count from 4.5–11 × 10^9^/L, and pseudo-platelets count is from 150–400 × 10^9^/L; at high concentrations (HH3), the human RBCs value is from 5.51–6.85 × 10^12^/L, pseudo-leucocytes count is from 11.1–50 × 10^9^/L, and pseudo-platelets count is from 401–600 × 10^9^/L in the optimum environment for each concentration level. Ten sets of samples (one set equals three concentrations) were measured to determine the assigned value (mean ± 2SD). Ten sets of samples were also sent simultaneously to two other laboratories with two models to be tested (Mindray BC 2000 and CD 1700) to determine the assigned values. The three devices used an impedance method to analysis samples. These laboratories have participated in the EQA programs, have obtained the quality standards, and have received good EQA results (the index evaluates the difference from the assigned value, z-score ≤ 1; the z-score results are taken from the results of the EQA statistics of the last 2 years performed by CSQL of HCMC). 

The composite sample, after determining the assigned value for the number of blood cell on each type of machine, was sent to 30 different laboratories for comparison to assess the impact of the actual environment on the sample. Selected laboratories had to meet the following criteria: full participation in the EQA programs for hematology and not marked with warning signs (z-score > 3). Hematology analysis was performed on the machines being studied: ABX Micros 60, Celldyn 1700, and Mindray BC 2000.

How the samples were sent: Each laboratory received a separate set of samples of three different concentrations (each set includes three vials) in cold shipping conditions (2 °C–8 °C) and the laboratory returned the result answer sheet (the results include three parameters at three different concentration levels) by post. The sending of samples and the returning of laboratory results are confidential information.

The results sent back were processed by Excel software and evaluated by the Levey–Jennings chart [9]. The graphs show the variation of each presumptive RBCs, WBCs, PLTs count around the assigned value for each model. Analysis results were considered a pass when the amounts of each laboratory’s presumptive human RBCs, pseudo-leucocytes, and pseudo-platelets count analysis within the range assigned value ± 2SD was greater than 80% (in this case, each laboratory had to provide 8/10). We compared the variation (CV%) of 10 sample analysis results from 10 laboratories with the allowable variation value of hematology analyzers and international standards.

We also evaluated the homogeneity of results (*n* = 10) for each type of analysis device and between different laboratories at the same concentration level, using two-way ANOVA analysis (statistically significant α = 0.05) and Excel software. Hypothesis H_0_ posed was: the results for the number of blood cells in the laboratories for each hematology analyzer are similar (hypothesis H_0_ by row) and the results of blood cell count analysis between different laboratories are the same (hypothesis H_0_ by column). If F_1_ (by row) < F_crit_ and F_2_ (by column) < F_crit_, accept hypothesis H_0_; otherwise, reject hypothesis H_0_.

### 2.4. Evaluation of Results between Laboratories

ANOVA (α = 0.05) by Excel 2010 software was used for the above calculations.

## 3. Results

### 3.1. Determination of Three Levels of High, Normal, and Low Concentration Values of Human RBCs, Pseudo-Leucocytes, and Pseudo-Platelets

Based on the clinical value threshold of three types of blood cell (RBCs, WBCs, and PLTs [10] along with the linear range of the three models, we investigated the ABX Micros 60, Celldyn 1700, and Mindray BC 200 according to the manufacturer’s announcement along with pathological high thresholds to determine the high, normal, and low thresholds. These threshold values are shown in the Table 1.

As shown in Table 1, the low level was identified according to the lowest levels of three hematology analyzers and clinical values. Meanwhile, the normal level was detected in the clinical value interval and the high level was established from the highest value of clinical threshold and pathological values. 

These levels are clearly shown below:-Low level: human RBCs from 0.23 to smaller 3.5 × 10^12^/L, pseudo-leucocytes from 0.8 to smaller 4.5 × 10^9^/L, pseudo-platelets from 10- to smaller 150 × 10^9^/L.-Normal level: human RBCs from 3.5–5.5 × 10^12^/L, pseudo-leucocytes from 4.5–11 × 10^9^/L, pseudo-platelets from 150–400 × 10^9^/L.-High level: human RBCs from larger 5.5 to 6.85 × 10^12^/L, pseudo-leucocytes from larger 11 to 50 × 10^9^/L, pseudo-platelets from larger 400 to 600 × 10^9^/L.

### 3.2. Evaluation of the Assigned Value

Samples of blood cells at three different concentrations (HH_1_: low, HH_2_: normal, HH_3_: high, 10 samples for each concentration) were run on the ABX Micros 60 series and then run on the Celldyn 1700 series and Mindray BC 2000 at two other different laboratories to define the assigned range (mean ± 2SD).

Statistical analysis results from Excel software determined the values of human RBCs, pseudo-leucocytes, and pseudo-platelets for the ABX Micros 60, Celldyn 1700, and Mindray BC2000 analyzers. The assigned value is shown in Table 2.

The results of running 10 samples on the same model to determine the assigned value for that model show that, for the ABX Micros 60, CellDyn 1700, and Mindray BC 2000 to analyze the three concentrations of high, normal, and low, CV% values of pseudo-leucocytes, human RBCs, and pseudo-platelets are smaller than allowable variation value of hematology analyzers and according to CLIA′88 standards and the standards of Japan Medical Association. CV% values shown in Table 3.

After the assigned value of each model was determined, composite samples with different concentrations were sent to 10 laboratories running the ABX Micros 60, Celldyn 1700, and Mindray BC 2000 analyzers (Appendix A).

The results of these 10 laboratories are presented on a Levey–Jennings chart to evaluate the variability in human RBCs values, pseudo-leucocytes, and pseudo-platelets values from the assigned value (mean ± 2SD), and were determined for the ABX Micros 60, Celldyn 1700, and Mindray BC 2000 analyzers. The data are shown in Figure 1.

At low concentrations, there are two values of pseudo-leucocytes on the ABX Micros 60 and one value of human RBCs on the Celldyn 1700 outside the mean ± 2SD range. At normal concentrations, there is one value of pseudo-leucocytes, one value of pseudo-platelets on the ABX Micros 60, one value of pseudo-leucocytes, and one value of pseudo-platelets on the Celldyn 1700 outside the mean ± 2SD range. At high concentrations, there is one value of pseudo-leucocytes, one value of human RBCs on the ABX Micros 60, and one value of human RBCs outside the mean ± 2SD range. On the Mindray BC2000, no values were outside the mean ± 2SD range for the three concentration levels. This proved that the assigned value and the results of the laboratories are reliable; the percentage of laboratories achieving analytical results matching the criteria set out was 80% on the ABX Micros 60 and Celldyn 1700 and 100% on the Mindray BC2000. Thus, hematological reference samples were qualified to be used to evaluate the results of laboratories using the ABX Micros 60, Celldyn 1700, and Mindray BC2000 analyzers.

We used two-way ANOVA analysis (statistically significant α = 0.05) and Excel software to evaluate the results among 10 laboratories using the electronic impedance method and results for the three different hematology analyzers at the same concentration level. The results of the statistical analysis are presented in Table 4.

At low concentrations, the results of human RBCs, pseudo-leucocytes, and pseudo-platelets on each analyzer were similar between laboratories, and the results of three analytical devices using the electronic impedance method were different, and this difference was statistically significant (*p* < 0.05).

At normal concentrations, the results of human RBCs, pseudo-leucocytes, and pseudo-platelets on each model between laboratories and the results of three analytical devices using the electronic impedance method were different, and this difference was statistically significant (*p* < 0.05).

At high concentrations, the results of RBCs, pseudo-leucocytes, and pseudo-platelets on each model were similar across 10 laboratories, and the results of three analytical devices using the electronic impedance method differed in the number of pseudo-leucocytes, pseudo-platelets, and this difference was statistically significant (*p* < 0.05), whereas the number of human RBCs was comparable.

## 4. Discussion

In previous research, we have fixed human RBCs, used goose and goat RBCs to replace human WBCs (pseudo-leucocytes), and used human PLTs (pseudo-platelets). The optimal formula was obtained using response surface methodology–central composite design (RSM-CCD); the results determined the optimal formulas for the blood cell of storage medium [8]. The sample life after optimization was determined to be 3 months. In this research, we performed threshold determination for three concentrations levels (high, normal, and low) in hematology analysis; at the same time, we determined the assigned values and evaluated the assigned values of the following hematology analyzers by impedance method: ABX Micros 60, Celldyn 1700, and Mindray BC 2000.

The analytical results from the different models will vary; thus, the results when analyzing the ABX Micros 60 series will be different from the results when evaluating the Celldyn 1700 and Mindray BC 2000 series. Therefore, a reference value is required for each machine model to be able to evaluate the results of the laboratory corresponding to the three models of ABX Micros 60, Celldyn 1700, and Mindray BC 2000 in the most objective manner. The closer the laboratory results are to the assigned values, the more accurate they are. The CV% value of the three blood cell parameters at the three control levels is smaller than the reference CV% according to the manufacturer’s publications and international standards. It indicates improved hematology quality control that is compatible with impedance analyzers.

The samples were then sent to 30 laboratories with the corresponding 3 models for testing. The analysis results showed that over 80% of the laboratories reached the consensus value. This proves that our samples can be used for the production of EQA hematological reference samples.

The results of the blood cell counts from the 10 laboratories at 3 concentrations were similar in each set. 

However, there were different results for the different analyzer models. This is possibly because of the different sensitivities, as well as the different reagents and construction of each model, cause the different results.

## 5. Conclusions

The study identified a range of values at three concentration levels (low, normal, and high) used in the preparation of hematology quality control samples. From that, the assigned values on the three impedance analyzers—ABX Micros 60, Celldyn 1700, and Mindray BC 2000—were established. The results of the assigned values for these three analyzers are used as a reference value for evaluate laboratories objectively.

The ABX Micros 60 and Celldyn 1700 series had 80% of laboratories with analytical results within the assigned value (mean ± 2SD). The Mindray BC 2000 series had 100% of laboratories with analytical results within assigned values. The measurement results for the number of human RBCs, pseudo-leucocytes, and pseudo-platelets on each analyzer were similar for each set between the ten laboratories, but the results of the three analyzers were different, and this difference was statistically significant (*p* < 0.05). Thus, the hematological reference samples that measured the number of blood cell has met the standards of EQA sample.

## Figures and Tables

**Figure 1 biomedicines-10-03169-f001:**
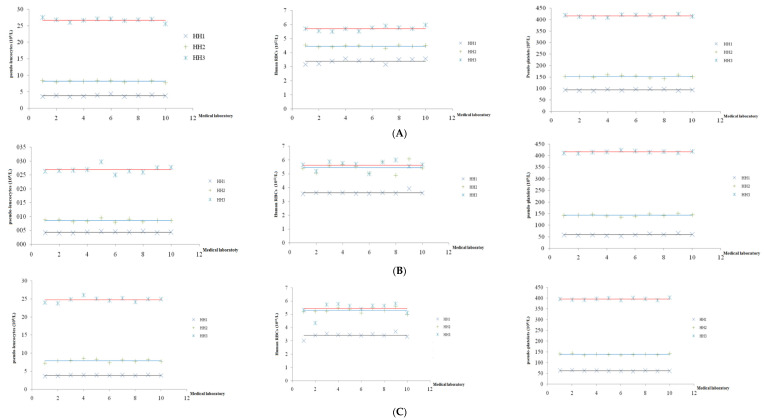
The results of the analysis of the assigned value at the concurrency levels of hematological reference samples run on the ABX Micros 60 (**A**), Celldyn 1700 (**B**) and Mindray BC 2000 (**C**). The three lines represent the mean (assigned value) for the low, normal, and high concentrations, respectively, and 10 points represent the results of 10 laboratories.

**Table 1 biomedicines-10-03169-t001:** High, normal, and low threshold values of three parameters: RBCs, WBCs, PLTs.

Parameters	Clinical Value [11]	Linear Range			Pathological Range
		ABX Micros 60	Celldyn 1700	Mindray BC 2000	
RBCs (10^12^ cells/L)	3.5–5.5	0.20–8.70	0.23–6.62	0.20–7.99	Smoker: 6.55–6.85 [12,13]
WBCs (10^9^ cells/L)	4.5–11	0.5–122.0	0.8–99.2	0.3–99.9	Acute leukemia: 20–50 [14]
PLTs (10^9^ cells/L)	150–400	10–2327	12–981	10–999	Essential thrombocythemia (ET): 400–600 [15]

**Table 2 biomedicines-10-03169-t002:** Assigned value of three concentration levels on three hematology analyzers using an electronic impedance method.

Parameters	ABX Micros 60			Celldyn 1700			Mindray BC 2000		
	HH1	HH2	HH3	HH1	HH2	HH3	HH1	HH2	HH3
RBCs (10^12^ cells/L)	3.43 ± 0.12	4.43 ± 0.14	5.70 ± 0.22	3.60 ± 0.12	5.52 ± 0.16	5.80 ± 0.20	3.43 ± 0.38	5.24 ± 0.39	5.43 ± 0.66
WBCs (10^9^ cells/L)	3.83 ± 0.14	8.32 ± 0.34	26.60 ± 0.66	4.33 ± 0.22	8.51 ± 0.40	26.41 ± 0.84	3.75 ± 0.28	7.88 ± 0.90	24.83 ± 1.35
PLTs (10^9^ cells/L)	95 ± 6.34	151 ± 6.98	416 ± 9.44	60 ± 7.10	143 ± 6.54	416 ± 8.26	63 ± 3.92	138 ± 6.26	396 ± 8.61

**Table 3 biomedicines-10-03169-t003:** CV% results of three concentration levels after analysis on the three hematology analyzers.

	ABX Micros 60			Celldyn 1700			Mindray BC 2000			Allowable CV%	
	HH1	HH2	HH3	HH1	HH2	HH3	HH1	HH2	HH3	Reference Value 1 * [16]	Reference Value 2 ** [17]
Pseudo-leucocytes	1.90	2.11	1.24	2.47	2.38	1.60	3.31	5.73	2.72	7.50	5.00
Human RBCs	1.07	1.59	1.89	1.56	1.40	1.70	5.57	3.73	6.08	3.00	4.00
Pseudo-platelets	3.35	2.31	1.13	5.90	2.28	0.99	3.13	2.27	1.09	12.50	7.00

* CLIA′88 standards ** Standards of Japan Medical Association.

**Table 4 biomedicines-10-03169-t004:** The results of statistical analysis between different laboratories.

	HH1					HH2					HH3				
	F_1_	F_crit_ _(row)_	F_2_	F_crit (colum)_	*p*-Value	F_1_	F_crit (row)_	F_2_	F_crit (colum)_	*p*-Value	F_1_	F_crit (row)_	F_2_	F_crit (colum)_	*p*-Value
Human RBCs	1.36	2.46	6.72	3.55	6.61 × 10^−3^	1.35	2.46	58.64	3.55	1.31 × 10^−8^	2.37	2.46	3.07	3.55	0.07
Pseudo-leucocytes	2.05	2.46	20.06	3.55	2.62 × 10^−5^	1.44	2.46	9.94	3.55	1.23 × 10^−3^	1.10	2.46	16.68	3.55	7.97 × 10^−5^
Pseudo-platelets	1.75	2.46	494.16	3.55	1.87 × 10^−16^	0.69	2.46	21.12	3.55	1.90 × 10^−5^	1.59	2.46	93.30	3.55	3.16 × 10^−10^

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
