# Peer review of "Determination of the Assigned Values of Blood Cells by an Impedance Method for Hematological Reference Samples Used in Hematology External Quality Assessment (EQA) Programs"

_biomedicines, 2022, doi:10.3390/biomedicines10123169_

Round 1

Reviewer 1 Report (Previous Reviewer 2)

This resubmission has improved considerably. The authors have addressed some of the previous comments. However, before the acceptance of this work the authors are encouraged to message the language in abstract and conclusion part, making them different instead of simply copy and paste. Abstract should focus on novelty and impact of the work, while the conclusion should contain the biggest findings.

Author Response

Dear Reviewers,

Thank you very much Dear Reviewers for all valuable comments to improve our manuscript. I have revised our manuscript as per Reviewer’s comments. In addition, the revised words are highlighted in yellow colour in the manuscript for your convenience.

I shall be pleased for your rapid handling of this revised manuscript and it will be greatly appreciated.

Thank you and best regards.

Sincerely,

Ngoc Nguyen Vo

Reviewer 2 Report (Previous Reviewer 1)

Minor typographical errors:  Table 1 - leukemia is mis spelled; table 2  - title should say three concentration levels (instead of level) along with other minor typographical errors.

The final paragraph of the discussion section does not quite make sense. I believe the intent is to say that while the 10 analyzer assessment confirmed the similar results on each set - the results between different analyzers models were different - as expected.  Please clarify/rewrite this sentence.

In the discussion section - a portion of the results related to CVs on each analyzer type are reported (second paragraph). Rather than including these results in this discussion section,  a summary in this section with would be sufficient  - with the actual CV values reported in the results section instead (a section in results on the 10 analyzer comparison with a table or list of CVs achieved could replace a portion of this discussion)

Author Response

Dear Reviewers,

Thank you very much Dear Reviewers for all valuable comments to improve our manuscript. I have revised our manuscript as per Reviewer’s comments. In addition, the revised words are highlighted in yellow colour in the manuscript for your convenience.

I shall be pleased for your rapid handling of this revised manuscript and it will be greatly appreciated.

Thank you and best regards.

Sincerely,

Ngoc Nguyen Vo

This manuscript is a resubmission of an earlier submission. The following is a list of the peer review reports and author responses from that submission.

Round 1

Reviewer 1 Report

Very interesting paper outlining the process for development of samples for external quality assessment. Please see attached file for specific suggestions.

Author Response

15th September, 2022

Submission of the Revised Manuscript – Ms ID: applsci-1308328

Title: “Determination of the assigned value of blood cell by impedance method of hematological reference samples used in hematology external quality assessment (EQA) programs”

Dear Reviewers,

Thank you very much Dear Reviewers for all valuable comments to improve our manuscript. I have revised our manuscript as per Reviewer’s comments. In addition, the revised words are highlighted in red color in the manuscript for your convenience.

I shall be pleased for your rapid handling of this revised manuscript and it will be greatly appreciated.

Thank you and best regards.

Sincerely,

Ngoc Nguyen Vo

Reviewer 2 Report

This manuscript by Nguyen Vo et al. aim to develop an external quality assessment (EQA) to check the quality of hematology tests for hematology laboratories in Vietnam. The hematological reference sample is controlled at 3 levels of low, normal and high concentrations, sent out for blind test, and the assigned value is determined by ABX Micros 60, Celldyn 1700, and Mindray BC 2000. Although the idea is encouraging, I found the experimental design needs significant improvement, and therefore a rejection decision is made.

1. The variations are limited in this study, e.g., the devices are sent out from the same place, and this greatly reduces the meaning of testing the samples at 10 different located lab; what is the protocol of the samples being differently processed in different locations? How would these different treatment affects the blood samples in principle?

2. In figures 2,3 middle panel, the RBCs at normal and high level have a similar determined value (HH2 vs. HH3), I for one will encourage authors to redetermine the optimized RBC level for sending out measurement. So the sample concentrations at three levels requires a redesign.

I also found that there are many abbreviations (e.g., RBC, WBC, PLT, RSM-CCD, etc.) without full name at the first appearance in the main text.

Author Response

(The authors gave the same response as above.)

Round 2

Reviewer 2 Report

The manuscript has improved and the authors have dealt with most of the reviewer comments satisfactorily so that I can recommend acceptance.